# Accuracy of Dynamic Computer-Assisted Implant Placement: A Systematic Review and Meta-Analysis of Clinical and In Vitro Studies

**DOI:** 10.3390/jcm10040704

**Published:** 2021-02-11

**Authors:** Sigmar Schnutenhaus, Cornelia Edelmann, Anne Knipper, Ralph G. Luthardt

**Affiliations:** 1Center for Dentistry, Dr. Schnutenhaus Community Health Center, 78247 Hilzingen, Germany; edelmann@schnutenhaus.de (C.E.); knipper@schnutenhaus.de (A.K.); 2Center of Dental, Oral and Maxillofacial Medicine, Clinic for Dental Prosthetics, Ulm University, 89081 Ulm, Germany; ralph.luthardt@uniklinik-ulm.de

**Keywords:** computer-assisted surgery, dynamic navigation, accuracy, prostheses and implants, dental implants, dental implantation, computer-aided surgery

## Abstract

The aim of this systematic review and meta-analysis is to analyze the accuracy of implant placement using computer-assisted dynamic navigation procedures. An electronic literature search was carried out, supplemented by a manual search. The literature search was completed in June 2020. The results of in vitro and clinical studies were recorded separately from each other. For inclusion in the review, the studies had to examine at least the prosthetically relevant parameters for angle deviation, as well as global deviation or lateral deviation at the platform of the implant. Sixteen of 320 articles were included in the investigation: nine in vitro and seven clinical studies. The meta-analysis showed values of 4.1° for the clinical studies (95% CI, 3.12–5.10) and 3.7° for the in vitro studies (95% CI, 2.31–5.10) in terms of the angle deviation. The global deviation at the implant apex of the implant was 1.00 mm for the clinical studies (95% CI, 0.83–1.16) and 0.91 mm for the in vitro studies (95% CI, 0.60–1.12). These values indicate no significant difference between the clinical and in vitro studies. The results of this systematic review show a clinical accuracy of dynamic computer-assisted navigation that is comparable to that of static navigation. However, the dynamic navigation systems show a great heterogeneity that must be taken into account. Moreover, currently there are few clinical data available. Therefore, further investigations into the practicability of dynamic navigation seem necessary.

## 1. Introduction

The objective of an implant prosthetic restoration is the functional and esthetic rehabilitation of the masticatory organ after tooth loss [1]. Prosthetically driven planning has been shown to be suitable for achieving this goal in an optimal and predictable way [2]. When planning implant positions, various aspects must be considered and assessed equally. For example, the bone condition [3], the soft tissue condition [4], the inter-implant distance [5], or the position of the [6] cement space must be taken into account in the planning. Accordingly, the long-term success of an implant restoration is determined by multiple factors [7].

An established process is digital three-dimensional (3D) planning. The actual condition of the alveolar bone is recorded using 3D imaging (computed tomography (CT)) or CBCT (cone beam computed tomography) and merged with the target situation of a digitized prosthetic planning goal [1]. Such digital implant planning can be implemented using computer-assisted procedures [8]. With static computer-assisted procedures, clinically sufficient accuracy can be achieved [9]. The use of drill templates for implants has been sufficiently studied [10], and the use of these templates will achieve predictable results [11]. Various procedures for static guided implant placement have been established. For example, templates are used in which only the pilot drilling is carried out in a guided manner until the implant has been completely prepared and placed using the template [12]. It has been shown, however, that every single step in the digital workflow can lead to inaccuracies [13]. The intraoral positioning and fixation of the templates also have a significant influence on the accuracy [13]. Various authors suspected that implants that are placed using mucosa-supported templates show greater deviations from the planned implant position to the achieved implant position than those placed using tooth-supported templates [10]. Raico Gallardo et al. were able to refute this claim in their meta-analysis [14]. However, major inaccuracies in implantation can usually be traced back to application errors and not to the process per se [15]. In particular, errors in the positioning of the drilling template are to be mentioned here. A disadvantage of full-guided static navigation is listed as the fact that no intraoperative, condition-related changes are possible [11]. The use of closed drilling templates can also lead to bone overheating due to the lack of access for cooling liquid [16].

In addition to static computer-assisted surgical procedures, dynamic procedures are also available [1]. The position of the instruments is recognized in real-time through optical tracking systems using defined markers [17]. The position of the instruments and the three-dimensional planning situation can thus be followed on a screen by the implant surgeon [18]. This method may also be used in other dental issues. For example, endodontic treatments can be performed dynamically [19]. These procedures were first introduced in a variety of mostly preclinical studies [20,21,22], but were not widely used in clinical practice due to their complexity and cost [1]. The further development of computer technology and the associated computer-aided methods have increased the use of dynamic navigation in clinical practice in recent years [1]. The advantages of dynamic navigation are that any implant systems can be used thanks to open-sourced systems and that the disadvantages of storing a static template do not exist, especially with a flapless procedure [1]. Another advantage of dynamic navigation is that the implant placement is carried out with visibility and thus remains controllable, and the plan can be modified intraoperatively during the operation [11]. Another advantage over the use of drill templates is that the procedure is possible even with limited vertical space [23]. However, the complexity of the surgical procedure requires sufficient training of the surgeon and team, and a learning curve for the procedure must be observed [24]. Further development of implant surgical procedures based on technologies of virtual reality and augmented reality is rewarded with an increase in the quality of care [25].

The objective of this review was to determine the accuracy with which the planned implant position can be implemented clinically or in vitro when using dynamic navigation systems. 

## 2. Materials and Methods

### 2.1. Search Strategy

This systematic review was designed in compliance with the Preferred Reporting Items for Systematic Reviews and Meta-Analyses (PRISMA) guidelines [26]. The accuracy of the implant placement was studied using dynamic computer-assisted surgical procedures. The study included partially edentulous and edentulous patients. The search, presentation, and evaluation were carried out differently according to clinical and in vitro examinations. The question posed was: “How much accuracy can be achieved clinically and on the model with dynamic computer-assisted implant placement?” The protocol of this systematic review was registered in the international database of prospectively registered systematic reviews in health and social care PROSPERO (CRD-No. 42020179128).

### 2.2. PICO Questions

The selection criteria are shown in Table 1.

The PICO questions when searching for clinical (P in vivo) and in vitro (P in vitro) studies were as follows:(P) Population in vivo: Edentulous and partially edentulous patients who require an implant-prosthetic restoration;(P) Population in vitro: Plastic models of edentulous or partially edentulous jaws;(I) Intervention: Implant placement with a dynamic computer-assisted surgical procedure;(C) Comparison: Results of the clinical and in vitro investigations; and(O) Outcome Acccuracy: Deviation between the planned and actual achieved implant position.

### 2.3. Search Strategy

An electronic search was carried out in the MEDLINE (PubMed), EMBASE via Ovid, and Cochrane Central Register of Controlled Trials via Ovid databases. The electronic search was completed on 30 June 2020.

The search term was: (((((((((((((dental implantation [MeSH Terms]) OR dental implant [MeSH Terms]) AND dental navigation) OR computer aided dental implant) OR three dimensional dental planning) OR 3D dental planning) OR computer assisted dental implant) OR guided dental implant placement) OR dental surgical template) OR dental guided surgery) OR dental surgical guide) OR guided dental implant placement) AND ((dynamic) OR (robot *)).

In addition, a manual search for relevant further literature was carried out on the basis of the bibliographies of the included studies. Only publications in English and German were considered.

### 2.4. Study Selection

Duplicates were sorted out prior to screening. Two reviewers (C.E. and A.K.) reviewed all titles and abstracts independently. If there were differences of opinion, these were discussed with a third reviewer (S.S.). If a clear decision was made after the discussion, the publication was selected accordingly. If, after the discussion and assessment, a reviewer continued to report doubts, the title was included in the next selection round. No kappa values were calculated. The selection was made according to the criteria described in Table 1.

### 2.5. Risk of Bias, Quality Assessment, and Interstudy Heterogeneity

A quality assessment and a risk of bias assessment of the included studies were carried out by two independent reviewers (C.E. and S.S.). Since no randomized clinical studies were available for the question of this review article, the quality assessment was carried out in a modified form. The basis for this was the tool for assessing risk of bias in the Cochrane Handbook for Systematic Reviews of Interventions and the matrix for quality assessment by Esposito et al. [27]. The assessment was based on the following points: Allocation concealment, sample size calculation, inclusion and exclusion criteria, incomplete outcome data, blinding of outcome assessment, selective reporting and appropriate statistical analysis, and other bias.

### 2.6. Data Extraction and Method of Analysis

The data were independently compiled in tables by the investigators. These tables contain the following parameters of the clinical examination: author(s), year of publication, study design (prospective/retrospective), number of patients, number of implants, edentulism, location of the implants, implant system, guide system, and planning software. The following parameters were included in the evaluation of the in vitro examination: author(s), year of publication, number of models, number of implants, edentulism, location of the implants, implant system, guide system, and planning software. The target variables were the deviation between the planned and the actual achieved implant position. The following were recorded in the tables: angle deviation, horizontal coronal and apical deviation, vertical coronal and apical deviation, and coronal and apical global deviation. 

The statistical analysis was carried out using the software program R Version 4.0.2 (The R Foundation for Statistical Computing Platform (R Foundation for Statistical Computing, Vienna, Austria.). The meta-analysis was carried out for the parameters of angle deviation and global deviation at the coronal end of the implant. In studies that had not published any information on the confidence interval (CI), the standard deviation (SD) value was used to calculate a CI equivalence. As there was evidence of heterogeneity between the included studies, totals were calculated using random-effects meta-analysis for continuous variables. The models were based on the variances approach of DerSimonian and Laird. The heterogeneity was assessed using Cochran’s Q test (*p* < 0.001 (CI 95%) and I2 statistic (I2 > 50%)). In addition, a descriptive analysis was carried out between the clinical and in vitro studies using the mean values. For this purpose, the angle deviation and the deviation at the coronal end of the implant were again used. A *p*-value of <0.05 was considered statistically significant. This descriptive analysis was performed with IBM SPSS® Statistics Version 27.0 (IBM Corp., Armonk, NY, USA).

## 3. Results

### 3.1. Study Selection

The study selection is described in the flow chart in Figure 1.

The electronic search yielded 320 hits, which were supplemented by 12 publications from the manual search. After screening the titles and then the abstracts, 47 publications were viewed as full texts. The review of the full texts led to the exclusion of a further 31 publications for the following reasons: deviation measurement with other parameters [22,28,29,30], does not analyze all deviations [31], zygomatic implants [32,33], different research question [17,34,35,36,37,38,39,40,41], overview article or literature review [1,11,24,42,43,44,45,46,47,48], and less then 10 patients/five models or less then 10 implants [23,49,50]. The inclusion criteria were met in nine publications of in vitro studies (Table 2) [20,21,51,52,53,54,55,56,57]. Clinical data could be used from seven publications that met the inclusion criteria (Table 3) [58,59,60,61,62,63,64].

### 3.2. Quality of the Studies

The studies were evaluated separately according to clinical and in vitro investigations in a methodical risk analysis. The risk assessment was carried out using established procedures and adapted to the samples being examined. Selection bias was included for completeness but is negligible in in vitro studies. The risk assessments are shown in Figure 2 and Figure 3.

A blinded assessment was clearly described in only a few studies [58,63]. In the other works, this was recorded as an increased risk factor. All studies showed a low risk rating for attrition and reporting bias. The financial participation of an industrial partner was assessed as a further possible risk factor [51,53]. There was no evidence of financial support for the other studies. 

### 3.3. Outcomes

In the nine in vitro studies, seven commercially available navigation systems and one prototype were examined. A total of 125 models were implanted. Five hundred and sixty-nine implants were placed and evaluated. Four commercially available navigation systems were used in the seven clinical studies. A total of 298 patients received implants. Seven hundred and fifty-seven implants were placed and evaluated. The navigation systems all showed a significantly different design, particularly in terms of the structure and the spatial distribution of the markers in the tracking systems for detecting the drill position. Different planning software was also used depending on the system, and various implant systems were used. 

In order to compare the planned implant position with the actual achieved implant position, a CBCT was performed postoperatively in all clinical publications. The evaluation was carried out after superimposing the pre- and post-operative CBCT and the planning data. The results of the accuracy tests of the in vitro studies are shown in Table 4 and those of the clinical studies in Table 5. 

Different parameters were recorded in the publications. With the inclusion criteria, the minimum requirement was defined as the angle deviation and the linear or global deviation at the coronal end of the implant. The representation in the forest plot diagrams and the descriptive comparison between clinical and in vitro examinations were therefore limited to these two prosthetically relevant target values. The angular deviation of the in vitro studies is shown in Figure 4. In the in vitro examinations, the global deviation was given in five papers and the linear coronal deviation of the implant in seven papers. For this reason, two forest plots were calculated. 

Figure 5 summarizes the global deviations at the implant platform. Figure 6 shows the global deviation at the implant apex. The following Figure 7 shows the lateral deviations at the implant platform without looking at the horizontal deviation. In the clinical studies, data on both the angular deviation (Figure 8) and the global deviation at the implant platform were available (Figure 9). Figure 10 summarized the global deviation at the implant apex. All of the forest plots confirmed a substantial heterogeneity in all of the parameters in both groups. The results of these analyses were *p* < 0.001 and *I*^2^ = 97.4%–99.6%.

#### 3.3.1. Coronal Deviation

The results of the meta-analysis showed comparable mean values. The global deviation in the clinical studies was 1.00 mm (95% CI, 0.83–1.16). In the in vitro studies, the global deviation was 0.91 mm (95% CI, 0.60–1.21) and the lateral deviation was 1.01 mm (95% CI, 0.68–1.34). The forest plots demonstrate that the accuracy was highly system-dependent. Both evaluation groups showed outliers for the mean accuracy. It can also be seen from the graph that the scatter around the mean values in the clinical examinations is higher.

#### 3.3.2. Apical Deviation

The results of the meta-analysis showed comparable mean values. The global deviation in the clinical studies was 1.33 mm (95% CI, 0.98–1.68). In the in vitro studies, this global deviation was 1.04 mm (95% CI, 0.76–1.33). The forest plots demonstrate that the accuracy was highly system-dependent. Both evaluation groups showed outliers for the mean accuracy. It can also be seen from the graph that the scatter around the mean values in the clinical examinations is higher.

#### 3.3.3. Angle Deviation

The results of the meta-analysis also showed comparable mean values (clinical 4.1° (95% CI, 3.14–5.10) and in vitro 3.7° (95% CI, 2.31–5.10)). The forest plots showed little scatter in both the clinical and in vitro studies.

The dynamic navigation also showed comparable accuracies compared to the static guided surgery. The mean values of this meta-analysis for dynamic navigation and the values from two [10,65] meta-analyses for static guided implant placement are shown in Table 6. This table is supplemented with the results of an examination [66] of the accuracy of freehand implants. Thus far, there has been no systematic review, as only a few studies have been published. To classify the in vitro studies on the clinical studies, we carried out a static analysis for their variance. This analysis should have a descriptive value, since there is not yet enough data on an actual one. No statistically significant differences in the achieved accuracy could be detected when comparing in vitro and clinical studies. For the angle deviation, the Mann–Whitney *U* test indicated a significance level of *p* = 0.964 and a deviation at the implant shoulder of *p* = 0.685.

## 4. Discussion

In the present meta-analysis, the accuracy between the planned and actually achieved implant positions was evaluated using dynamic computer-assisted navigation. In their study design, all examinations showed clearly different influencing factors. Various navigation systems with fundamental differences in the structure of the optical tracking system and the arrangement of the markers were used. Furthermore, different implant planning programs and different implants were used. 

The high degree of heterogeneity of the measured parameters suggests that the evaluated studies examined navigation systems with significantly different qualities. The structure and arrangement of the marker structures seems to lead to different results. For reviews of computer-assisted static guided implants, there was a lower heterogeneity [65]. Moreover, the present study could not make any statement on clinical suitability for practice.

In this meta-analysis, in vitro and clinical studies were handled separately. This was based on clinical experience and the assumption that intraoral implementation in the patient is difficult. The opening of the mouth, movements of the patient, or the restricted view of the operating field can all have an influence [67]. Therefore, in this meta-analysis, it was hypothesized that there would be a significant difference between these two study designs. This was supported by a systematic review of static navigation by Bover-Ramos et al., in which significant differences between in vitro and clinical studies were found [65]. There were differences in the horizontal apical deviation and the angle deviation, and this statement was supported by Schneider’s meta-analysis [68]. Significant differences could not be detected in the evaluation of the dynamic navigation presented here. An evaluation of the subgroups in which in vitro and clinical studies with a similar study design—particularly the use of only one navigation system—are considered could not be carried out due to the as yet low number of studies. Therefore, a final assessment that the in vitro-achieved accuracy can also actually be achieved clinically cannot be made. The included studies showed moderate limitations in methodological quality and were thus in the range of the usual quality requirements [14]. In all clinical studies, the accuracy was determined by overlaying post-operative CBCTs. This should be assessed critically in further studies for considerations of radiation protection, since non-radiological examination methods are also available [15].

In the present evaluation, the focus was on the angle deviation and the deviation at the implant platform. This was justified in the prosthetic importance of these parameters. Inclined implant axes make it difficult to correctly design the approximal contacts. In their meta-analysis, Omori et al. found that after one year of follow-up, implants supporting angulated abutments yielded significantly more marginal bone loss than those supporting straight abutments [69]. The coronal position of the implant in particular has a decisive influence on the esthetic result [70]. Since our working group evaluated computer-assisted implants mainly as a primary prosthetic tool and only secondarily as a surgical tool, the values at the apex of the implant were not evaluated separately. Various authors have argued that this level of precision is necessary when assessing possible damage to relevant anatomical structures, e.g., the inferior alveolar nerves or the maxillary sinus [65]. An assessment of the lateral deviation at the implant tip makes only limited sense, however, since this value is directly dependent on the length of the implant used. However, in order to evaluate the possible risk of injury-sensitive anatomical structures such as the sinus, the mental foramen, and the mandible canal, these values were also evaluated. The dependence of the deviation on length is a purely geometric function and therefore has no clinical justification. Minimum distances to vulnerable anatomical structures must be observed in all procedures for guided implant placement [71]. The safety clearances given in the literature are between 1.0 [72] and 0.5 mm horizontally [73] and 1.7 [65] and 1.2 mm vertically [73]. Maintaining the horizontal position should be more reliable with a dynamic system that can be checked visually than with static systems in which a template covers the operating field.

The present evaluation also showed that the accuracy depends heavily on the navigation system. The considerable differences of the technique used was demonstrated by the work of Kang et al. The mean values in the work by Kang et al. showed significant deviations from the mean values calculated in the meta-analysis in the group of in vitro studies [57]. Overall, the studies showed a high heterogeneity of the data. A similar picture emerged in the clinical studies in the work of Pellegrino et al., as well as that of Aydemir and Arisan [61,63]. There were also considerable differences in the spread of the values in the individual studies. The influence of the planning software cannot be determined from the available data. Static navigation studies have shown that the accuracy of superimposing 3D datasets in software programs is different. For example there are significant differences in the accuracy of the matching of Standard Tessellation Language (STL) data and Digital Imaging and Communication in Medicine (DICOM) data [74]. All of the limitations and co-factors mentioned for static navigation, which influence the accuracy [10], can be transferred to the planning programs used here.

The results of this meta-analysis are in the range of the results obtained from meta-analyses for static navigation [10,65,67]. Further investigations must show whether this also applies to dynamic navigation systems that have not been investigated so far. There are also only studies by one working group of individual systems. Therefore, there are no statements about the robustness of the values when the systems are used by different surgeons. In particular, in addition to the accuracy, the feasibility in everyday clinical practice must also be examined in further studies. The parameters to be considered here are the time required, the surgical difficulty in use, and, last but not least, the costs.

## 5. Conclusions

With dynamic computer-assisted navigation, mean deviations between the planned and actually achieved implant position of 1.00 mm clinically and 0.91 mm in vitro were calculated at the platform of the implant. The mean global deviation at the implant apex was 1.33 mm clinically and 1.04 mm in vitro. The angle deviation averaged 4.1° in clinical examinations and 3.7° in in vitro examinations. These values were significantly different. These results of dynamic navigation show similar values as described in various systematic reviews for static navigation. The usual surgical safety distances must therefore also be observed with dynamic navigation. With further studies, recommendations can possibly be given for areas of application in which static and dynamic processes have advantages.

## Figures and Tables

**Figure 1 jcm-10-00704-f001:**
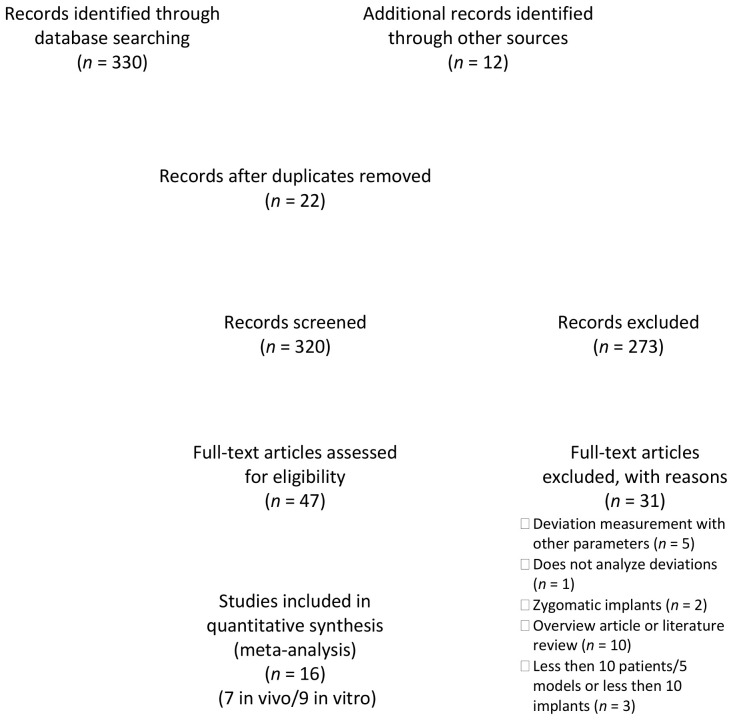
PRISMA flow diagram for the search strategy and selection process for the included studies.

**Figure 2 jcm-10-00704-f002:**
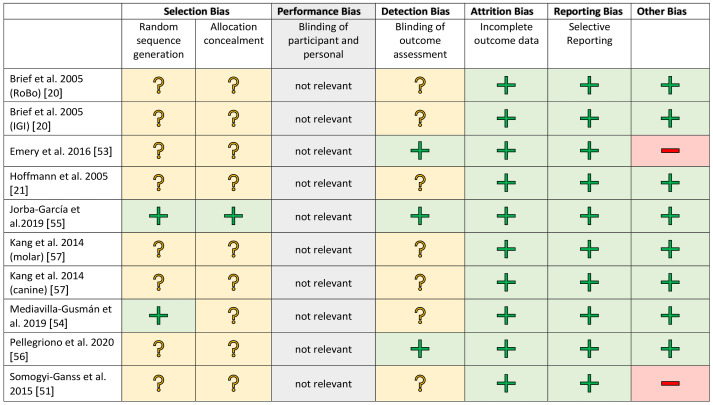
Risk of bias graph for the in vitro studies. +/green: low risk of bias; ?/yellow: unclear risk of bias; −/red: high risk of bias.

**Figure 3 jcm-10-00704-f003:**
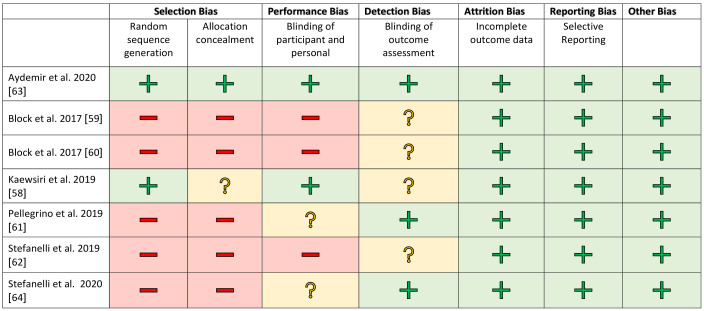
Risk of bias graph for the clinical studies. +/green: low risk of bias; ?/yellow: unclear risk of bias; −/red: high risk of bias.

**Figure 4 jcm-10-00704-f004:**
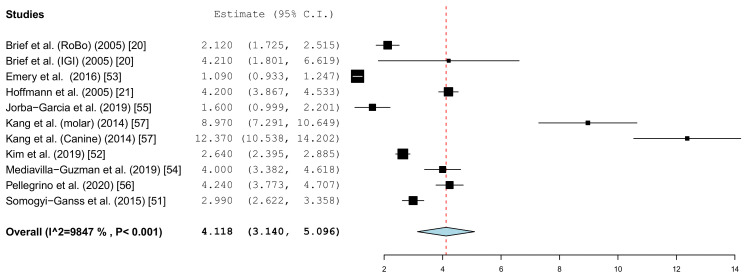
Forest plot demonstrating the angular deviation (°) for all of the selected in vitro articles.

**Figure 5 jcm-10-00704-f005:**
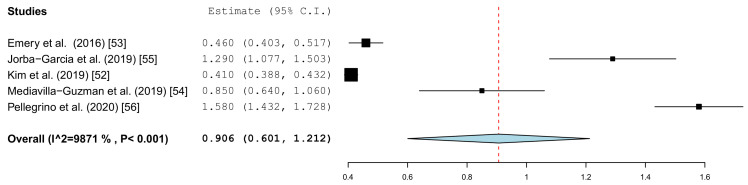
Forest plot demonstrating the global deviation (mm) at the implant platform measured for all of the selected in vitro articles.

**Figure 6 jcm-10-00704-f006:**
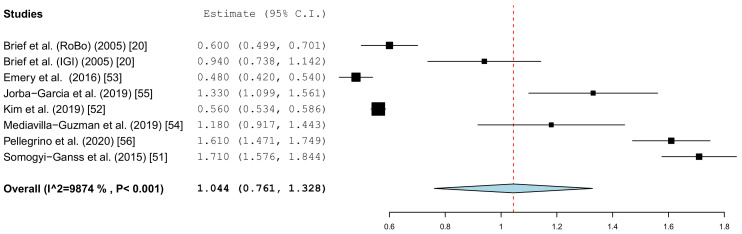
Forest plot demonstrating the global deviation (mm) at the implant apex measured for all of the selected in vitro articles.

**Figure 7 jcm-10-00704-f007:**
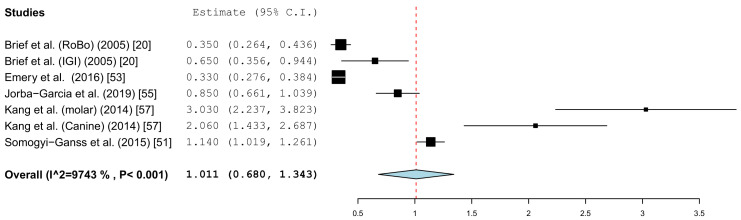
Forest plot demonstrating linear lateral deviation (mm) at the implant platform measured for all of the selected in vitro articles.

**Figure 8 jcm-10-00704-f008:**
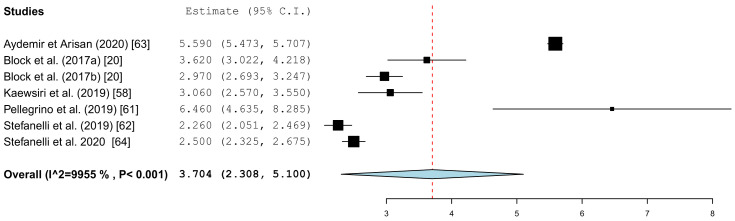
Forest plot demonstrating the angular deviation (°) for all of the selected clinical articles.

**Figure 9 jcm-10-00704-f009:**
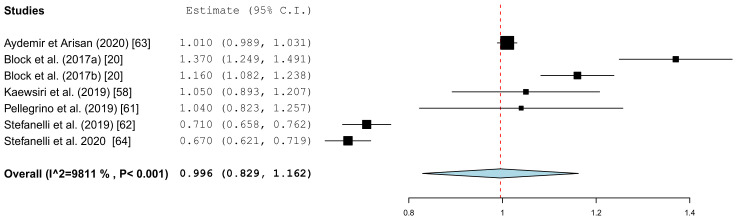
Forest plot demonstrating the global deviation (mm) at the implant platform measured for all of the selected clinical articles.

**Figure 10 jcm-10-00704-f010:**
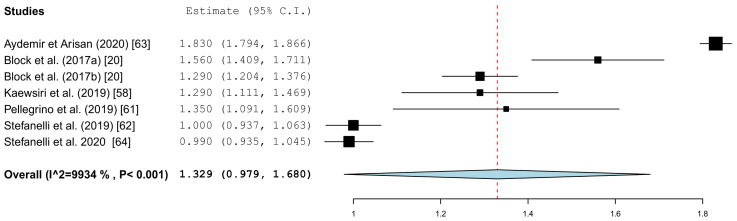
Forest plot demonstrating the global deviation (mm) at the implant apex measured for all of the selected clinical articles.

**Table 1 jcm-10-00704-t001:** Selection criteria.

	Clinical Studies	In Vitro Studies
Inclusion criteria	Clinical study Partially edentulous or edentulous jaws At least 10 patients Implant placement with a dynamic computer-assisted surgical procedure Studies with an outcome accuracy between the planned and actual achieved implant position At minimum, the following parameters had to be recorded: angle deviation and linear or global deviation at the implant platform of the implant.	Models (plastic) of edentulous or partially edentulous jaws At least 5 models with a total of more than 10 implants Implant placement with a dynamic computer-assisted surgical procedure Studies with an outcome accuracy between the planned and actual achieved implant position At minimum, the following parameters had to be recorded: angle deviation and linear or global deviation at the implant platform of the implant.
Exclusion criteria	Only static guided implant placement Cadaver or animal studies Expert opinions Unclear description of the procedure Zygomatic, pterygoid, and orthodontic implants Multiple publications from the same patient	Only static guided implant placement Cadaver or animal studies Expert opinions Unclear description of the procedure Zygomatic, pterygoid, and orthodontic implants Multiple publications of the same model series

**Table 2 jcm-10-00704-t002:** Characteristics of the selected in vitro studies included in the review.

Study	Number of Models	Number of Implants	Edentulism	Jaw	Implant System	Guide System	Planning Software
Brief et al. (2005) [20] (Robo)	5	15	Partially	Mandible	NR	RoboDent; RoboDent, GmbH, Berlin, Germany	RoboDent; RoboDent, GmbH, Berlin, Germany
Brief et al. (2005) (IGI) [20]	5	15	Partially	Mandible	NR	IGI DenX; Denx Ltd., Moshav Ora, Jerusalem, Israel	IGI DenX; Denx Ltd., Moshav Ora, Jerusalem, Israel
Emery et al. (2016) [53]	27	47	Partially and fully edentulous	Maxilla and mandible	Zimmer/Biomet 3i, Palm Beach, FL, USA	X-Guide; X-Nav Technologies, LLC, Lansdale, PA, USA	X-Guide; X-Nav Technologies, LLC, Lansdale, PA, USA
Hoffmann et al. (2005) [21]	16	112	Fully edentulous	Mandible	NR	Vector-Vision Compact; VVC, BrainLAB, Heimstetten, Germany	NR
Kang et al. (2014) (molar) [57]	10	20	Fully edentulous	Mandible molar region	Dentium implant Fx4314; Dentium, Seoul, Korea	CBYON suite system; CBYON Inc., Mountain View, CA, USA	SimPlant; Materialise Dental, Leuven, Belgium
Kang et al. (2014) (canine) [57]	10	20	Fully edentulous	Mandible canine region	Dentium implant Fx4314; Dentium, Seoul, Korea	CBYON suite system; CBYON Inc., Mountain View, CA, USA	SimPlant; Materialise Dental, Leuven, Belgium
Kim et al. (2015) [52]	10	110	Partially	Maxilla and mandible	Ostem TS; Osstem Implant, Seoul, Korea	Polaris Vicar; Northern Digital Inc., Waterloo, ON, Canada	InVivoDental; Anatomage, San Jose, CA, USA
Mediavilla-Guzmán et al. (2019) [54]	10	20	Partially	Maxilla	BioHorizons, Birmingham, AL, USA	Navident System; ClaroNav Inc., Toronto, ON, Canada	Navident System; ClaroNav Inc., Toronto, ON, Canada
Jorba-García et al. (2019) [55]	6	18	Partially	Mandible	Ticare In-Hex; MG Mozo-Grau, Valladolid, Spain	Navident System; ClaroNav Inc., Toronto, ON, Canada	Navident System; ClaroNav Inc., Toronto, ON, Canada
Pellegrino et al. (2020) [56]	16	112	Edentulous	Maxilla	Southern Implants, Irene, South Africa	ImplaNav; BresMedical, Sydney, Australia	ImplaNav; BresMedical, Sydney, Australia
Somogyi-Ganss et al. (2015) [51]	10	80	Partially	Maxilla and mandible	NR	Prototype Navident; Claron Technology Inc., Toronto, ON, Canada	Prototype Navident; Claron Technology Inc., Toronto, ON, Canada

NR: not reported.

**Table 3 jcm-10-00704-t003:** Characteristics of the selected clinical studies included in the review.

Study	Design	Number of Patients	Number of Implants	Edentulism	Jaw	Implant System	Guide System	Planning Software
Aydemir and Arisan (2020) [63]	Prospective	30	43	Partially	Maxilla	Southern Implants, Irene, South Africa	Navident System; ClaroNav Inc., Toronto, ON, Canada	Navident System; ClaroNav Inc., Toronto, ON, Canada
Block et al. (2017a) [59]	Prospective	80	80	Partially	Maxilla and mandible	NR	X-Guide; X-Nav Technologies, LLC, Lansdale, PA, USA	X-Guide; X-Nav Technologies, LLC, Lansdale, PA, USA
Block et al. (2017b) [60]	Prospective	NR (but more than 10)	219	Partially	Maxilla and mandible	NR	X-Guide; X-Nav Technologies, LLC, Lansdale, PA, USA	X-Guide; X-Nav Technologies, LLC, Lansdale, PA, USA
Kaewsiri et al. (2019) [58]	Prospective	30	30	Partially	Maxilla and mandible	Straumann Bone level (18), Straumann Bone Level Taper (9), Straumann Tissue level (3)	IRIS-100; EPED Inc., Kaohsiung City, Taiwan	IRIS-100; EPED Inc., Kaohsiung City, Taiwan
Pellegrino et al. (2019) [61]	Prospective	10	18	Partially and fully edentulous	Maxilla and mandible	Southern Implants IBT (16), Co-axis (2), Southern Implants, Irene, South Africa	ImplaNav; BresMedical, Sydney, Australia	ImplaNav; BresMedical, Sydney, Australia
Stefanelli et al. (2019) [62]	Retrospective	89	231	Partially and fully edentulous	Maxilla and mandible	NR	Navident System; ClaroNav Inc., Toronto, ON, Canada	Navident System; ClaroNav Inc., Toronto, ON, Canada
Stefanelli et al. (2020) [64]	Retrospective	59	136	Partially	Maxilla and mandible	Osseotite Tapered, Zimmer/Biomet 3i, Palm Beach, FL, USA	Navident System; ClaroNav Inc., Toronto, ON, Canada	Navident System; ClaroNav Inc., Toronto, ON, Canada

NR: not reported.

**Table 4 jcm-10-00704-t004:** Summary of the accuracies of the clinical studies.

Study	Year	Number of Patients	Number of Implants	Angle Deviation	SD	95% CI	Global Deviation at the Implant Platform	SD	95% CI	Linear Lateral Deviation at the Implant Platform	SD	95% CI	Vertical Deviation at the Implant Platform	SD	95% CI	Global Deviation at the Apex	SD	95% CI	Linear Lateral Deviation at the Apex	SD	95% CI	Vertical Deviation at the Apex	SD	95% CI
Aydemir and Arisan [63]	2020	30	43	5.59	0.39	4.87–6.42	1.01	0.07	0.87–1.18							1.83	0.12	1.60–2.10						
Block et al. [59]	2017a	80	80	3.62	2.73		1.37	0.55		0.87	0.42		0.93	0.60		1.56	0.69		1.09	0.66		0.96	0.66	
Block et al. [60]	2017b	NA > 10	219	2.97	2.09	2.46–3.44	1.16	0.59	1.03–1.27	0.74	0.43	0.60–0.78	0.76	0.60	0.68–0.94	1.29	0.65	1.14–1.43	0.9	0.55	0.73–0.99	0.78	0.6	0.68–0.94
Kaewsiri et al. [58]	2019	30	30	3.06	1.37	2.54–3.57	1.05	0.44	0.89–1.21							1.29	0.5	1.10–1.48						
Pellegrino et al. [61]	2019	10	18	6.46	3.95		1.04	0.47					0.43	0.34		1.35	0.56							
Stefanelli et al. [62]	2019	89	231	2.26	1.62		0.71	0.40								1.00	0.49							
Stefanelli et al. [64]	2020	59	136	2.50	1.04		0.67	0.29								0.99	0.33					0.55	0.25	

CI: confidence interval; NA > 10: not available, but more than 10 patients; SD: standard deviation.

**Table 5 jcm-10-00704-t005:** Summary of the accuracy of the in vitro studies.

Study	Year	Number of Models	Number of Implants	Angle Deviation	SD	Global Deviation at the Implant Platform	SD	Linear Lateral Deviation at the Implant Platform	SD	Vertical Deviation at the Implant Platform	SD	Global Deviation at the Apex	SD	Linear Lateral Deviation at the Apex	SD	Vertical Deviation at the Apex	SD
Brief et al. [20]	2005 (RoBo)	5	15	2.12	0.78			0.35	0.17			0.60	0.20	0.47	0.18	0.32	0.21
Brief et al. [20]	2005 (IGI)	5	15	4.21	4.76			0.65	0.58			0.94	0.40	0.68	0.31	0.61	0.36
Emery et al. [53]	2016	27	47	1.09	0.55	0.46	0.20	0.33	0.19	0.26	0.19	0.48	0.21	0.36	0.20	0.25	0.19
Hoffmann et al. [21]	2005	16	112	4.20	1.80												
Jorba-García et al. [55]	2019	6	18	1.60	1.30	1.29	0.46	0.85	0.41			1.33	0.50			0.88	0.47
Kang et al. [57]	2014 (molar)	10	20	8.97	3.83			3.03	1.81	0.76	0.84			2.76	1.03	1.96	0.93
Kang et al. [57]	2014 (canine)	10	20	12.37	4.18			2.06	1.43	1.14	1.25			3.31	2.07	1.42	1.01
Kim et al. [52]	2015	10	110	2.64	1.31	0.41	0.12					0.56	0.14				
Mediavilla-Guzmán et al. [54]	2019	10	20	4.00	1.41	0.85	0.48					1.18	0.60				
Pelleriono et al. [56]	2020	16	112	4.24	2.52	1.58	0.80			0.75	0.74	1.61	0.75			0.70	0.67
Somogyi-Ganss et al. [51]	2015	10	80	2.99	1.68			1.14	0.55			1.71	0.61	1.18	0.56	1.04	0.71

**Table 6 jcm-10-00704-t006:** Comparison the data from the present study of the angle deviation and the global deviation at the coronal end of the implants between dynamic and static navigation and a work on freehand implant placement (mean value and standard error (if specified)).

Study	Angle Deviation	Global Coronal Deviation	Global Apical Deviation
	Mean	Mean	Mean
Dynamic navigation in vitro	4.1°	1.03 mm	1.04 mm
Dynamic navigation clinical	3.7°	1.00 mm	1.33 mm
Static navigation—clinical review 1 [65]	3.6°	1.10 mm	1.40 mm
Static navigation—clinical review 2 [10]	3.5°	1.2 mm	1.40 mm
Freehand implant placement [66]	9.9°	2.77 mm	2.91 mm

## Data Availability

Not applicable.

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
