# Peer review of "Accuracy of Dynamic Computer-Assisted Implant Placement: A Systematic Review and Meta-Analysis of Clinical and In Vitro Studies"

_jcm, 2021, doi:10.3390/jcm10040704_

Round 1
Reviewer 1 Report
ABSTRACT: Please provide a simplified version of the abstract core in order to make it appealing to reader.
KEYWORDS: Please provide more keywords using MeSH terms to ensure a properly research in the medical database.
INTRODUCTION:
- In addition to static computer-assisted surgical procedures, dynamic procedures are also available (1). The position of the instruments is recognized in real-time through opti- cal tracking systems using defined markers (16). The position of the instruments and the three-dimensional planning situation can thus be followed on a screen by the implant surgeon. In order to better justify the authors findings regarding the above paragraph, it suggested to cite the following article: Gambarini, G., Galli, M., Stefanelli, L. V., Di Nardo, D., Morese, A., Seracchiani, M., De Angelis, F., Di Carlo, S., & Testarelli, L. (2019). Endodontic Microsurgery Using Dynamic Navigation System: A Case Report. Journal of endodontics, 45(11)
MATERIALS AND METHODS:
- The search term was: ((((((((((((((((dental implantation [MeSH Terms]) OR dental implant [MeSH Terms]) AND dental navigation) OR computer aimed dental implant) OR three dimensional dental planning) OR 3D dental planning) OR computer assisted dental im- plant) OR guided dental implant placement) OR dental surgical template) OR dental guided surgery) OR dental surgical guide) OR guided dental implant placemen t) AND ((dynamic) OR ( robot *)). Please correct any kind of typing errors in the paragraph above.
- Please before introducing any type of abbreviations it is recommended to properly explain them in order to help the reader following the development of the manuscript.
RESULTS
- Different parameters were recorded in the publications. With the inclusion criteria, the minimum requirement was defined as the angle deviation and the linear or 3D devia- tion at the coronal end of the implant. The representation in the forest plot diagrams and the descriptive comparison between clinical and in vitro examinations is therefore limited to these two prosthetically relevant target values (Figures 4-8). In the in vitro examina- tions, the global 3D deviation was given in 5 papers and the linear deviation at the exit point of the implant in 7 papers. For this reason, two forest plots were calculated (Figure 5 and 6). Please revise the above paragraph in order to provide a better explanation for Figures from 4 to 8.
- The studies were evaluated separately according to clinical and in vitro investiga- tions in a methodical risk analysis. The risk assessment was carried out using established procedures and adapted to the samples being examined. Selection bias was included for completeness, but is negligible in in vitro studies. The risk assessments are shown in Fig- ures 2 and 3. Please implement the above paragraph in order to better describe the “methodical risk-analysis” that guided the authors editing respectively Figures 2-3.
DISCUSSION:
- An assess- ment of the lateral deviation at the implant tip makes only limited sense, however, since this value is directly dependent on the length of the implant used. The following sentence is confused, please revise it.
- The mean values in the work by Kang et al showed significant deviations from the mean value calculated in the meta-analysis in the group of in vitro studies (55). The above sentence is unclear, please rearrange it in order to make it clearer.
- However, differences from studies on static navigation are listed here, for example in the accuracy of the matching of STL (Standard Tesselation Language) data and DICOM (Digital Imaging and Communication in Medicine) data (73). The present sentence is unclear, please revise it.
The systematic review provided by the authors is a work of a great interest for the journal. However, it is the revisor idea that some changes should be carried out in order to simplify some important passages. Therefore the authors are invited to carry out these major revisions outlined by the reviewer.
Author Response
The systematic review provided by the authors is a work of a great interest for the journal. However, it is the revisor idea that some changes should be carried out in order to simplify some important passages. Therefore the authors are invited to carry out these major revisions outlined by the reviewer.
We would like to thank you for the assessment that the manuscript is a work of interest for the Journal of Clinical Medicine.
We have carefully reviewed the reviewer´s advice and incorporated inspired changes.
Since Reviewer 2 has shown a revision of the translation into English, we have had this done by the MDPI English editing service.
We hope that the manuscript will now have a quality that it can be published in the Journal of Clinical Medicine.
Response to Reviewer 1
ABSTRACT: Please provide a simplified version of the abstract core in order to make it appealing to reader.
Simplifying the text is difficult for us here without losing important information from our point of view.
We have made two cuts. These explanations are adequately described in the main text.
KEYWORDS: Please provide more keywords using MeSH terms to ensure a properly research in the medical database.
Thank you for that comment. We added 4 MeSH terms:
„prostheses and implants; dental implants; dental implantation; computer-aided surgery”
INTRODUCTION:
In addition to static computer-assisted surgical procedures, dynamic procedures are also available (1). The position of the instruments is recognized in real-time through opti- cal tracking systems using defined markers (16). The position of the instruments and the three-dimensional planning situation can thus be followed on a screen by the implant surgeon. In order to better justify the authors findings regarding the above paragraph, it suggested to cite the following article: Gambarini, G., Galli, M., Stefanelli, L. V., Di Nardo, D., Morese, A., Seracchiani, M., De Angelis, F., Di Carlo, S., & Testarelli, L. (2019). Endodontic Microsurgery Using Dynamic Navigation System: A Case Report. Journal of endodontics, 45(11)
That is a good suggestion. Thus, it becomes plausible to the reader that dynamic processes also find applications outside of dental implantology. We have therefore added a sentence:
“This method may also be used in other dental issues. For example, endodontic treatments can be performed dynamically. (Gambarini et al.)”
MATERIALS AND METHODS:
The search term was: ((((((((((((((((dental implantation [MeSH Terms]) OR dental implant [MeSH Terms]) AND dental navigation) OR computer aimed dental implant) OR three dimensional dental planning) OR 3D dental planning) OR computer assisted dental im- plant) OR guided dental implant placement) OR dental surgical template) OR dental guided surgery) OR dental surgical guide) OR guided dental implant placemen t) AND ((dynamic) OR ( robot *)). Please correct any kind of typing errors in the paragraph above.
Thank you for this comment. We have corrected the typos.
Please before introducing any type of abbreviations it is recommended to properly explain them in order to help the reader following the development of the manuscript.
Thank you for this note. We have carefully passed through the manuscript again and have added previously unexplained abbreviations. We hope not to have missed a shortcut.
RESULTS
- Different parameters were recorded in the publications. With the inclusion criteria, the minimum requirement was defined as the angle deviation and the linear or 3D devia- tion at the coronal end of the implant. The representation in the forest plot diagrams and the descriptive comparison between clinical and in vitro examinations is therefore limited to these two prosthetically relevant target values (Figures 4-8). In the in vitro examina- tions, the global 3D deviation was given in 5 papers and the linear deviation at the exit point of the implant in 7 papers. For this reason, two forest plots were calculated (Figure 5 and 6). Please revise the above paragraph in order to provide a better explanation for Figures from 4 to 8.
Thank you for this important point. We have described the assignment of the individual forest plots in more detail and hopefully made the text clearer.
The paragraph now reads:
“Different parameters were recorded in the publications. With the inclusion criteria, the minimum requirement was defined as the angle deviation and the linear or 3D deviation at the coronal end of the implant. The representation in the forest plot diagrams and the descriptive comparison between clinical and in vitro examinations is therefore limited to these two prosthetically relevant target values. The angular deviation of the in vitro studies is shown in Figure 4. In the in vitro examinations, the global 3D deviation was given in 5 papers and the linear deviation at the exit point of the implant in 7 papers. For this reason, two forest plots were calculated. Figure 5 summarizes the 3D-deviations. The following figure 6 shows the 2D lateral deviations without looking at the horizontal deviation. In the clinical studies, data on both angular deviation (Figure 7) and 3-D deviation at the implant exit point were available. (Figure (8)
All forest plots confirmed a substantial heterogeneity in all parameter in both groups. The results of these analyses were P < 0.001 and I2 97.4% - 99.6%.”
- The studies were evaluated separately according to clinical and in vitro investiga- tions in a methodical risk analysis. The risk assessment was carried out using established procedures and adapted to the samples being examined. Selection bias was included for completeness, but is negligible in in vitro studies. The risk assessments are shown in Fig- ures 2 and 3. Please implement the above paragraph in order to better describe the “methodical risk-analysis” that guided the authors editing respectively Figures 2-3.
We did not fully understand the statement of this reference. We have therefore explained in more detail the risk assessment of the in vitro studies. We believe that the methodological aspect is now more recognisable.
„The studies were evaluated separately according to clinical and in vitro investigations in a methodical risk analysis. The risk assessment was carried out using established procedures and adapted to the samples being examined. Selection bias was included for completeness but is negligible in in vitro studies. The focus of a methodological risk assessment is therefore in the in vitro studies on the traceability of the evaluation and the static processing. In addition, "other bias" assessed the sponsorship of the studies. The risk assessments are shown in Figures 2 and 3.”
DISCUSSION:
An assess- ment of the lateral deviation at the implant tip makes only limited sense, however, since this value is directly dependent on the length of the implant used. The following sentence is confused, please revise it.
Thank you for this comment. We have added a sentence.
“The dependence of the deviation on length is a purely geometric function and therefore has no clinical justification.”
- The mean values in the work by Kang et al showed significant deviations from the mean value calculated in the meta-analysis in the group of in vitro studies (55). The above sentence is unclear, please rearrange it in order to make it clearer.
We have extended this passage and hopefully described it moreclearly:
“The present evaluation also showed that the accuracy depends heavily on the navigation system. The considerable differences of the technique used can be demonstrated by the work of Kang et al.. The mean values in the work by Kang et al showed significant deviations from the mean value calculated in the meta-analysis in the group of in vitro studies (56). Overall, the studies showed a high heterogeneity of the data“
However, differences from studies on static navigation are listed here, for example in the accuracy of the matching of STL (Standard Tesselation Language) data and DICOM (Digital Imaging and Communication in Medicine) data (73). The present sentence is unclear, please revise it.
That is a valuable comment. The sentence is really unclear. Here, information was also lost through translation. We have reworded it:
„The influence of the planning software cannot be determined from the available data. Static navigation studies have shown that the accuracy of superimposing 3D datasets in software programs is different. For example there are significant differences in the accuracy of the matching of STL (Standard Tesselation Language) data and DICOM (Digital Imaging and Communication in Medicine) data (74)“
Reviewer 2 Report
This article systematically reviewed the previous researches investigating the accuracy of dynamic navigation dental implant surgery and made a meta-analysis of the pooled data. Overall, the topic lacks significance and there are some major flaws in the methodology. The detailed comments are as follows.
- For the topic, the accuracy of the dental navigation system itself is not a question of much clinical significance. It’s more valuable to investigate whether dynamic navigation is superior to static computer guide systems or freehand in terms of accuracy, cost-effect, or patient-centered outcomes. Most of the included studies compared dynamic navigation with static guide or freehand. However, the author did not involve any analysis regarding those important comparisons in this systematic review.
- In the introduction section: “Implants that are placed using mucosa-supported templates show greater deviations from the planned implant position to the achieved implant position than those that were placed using tooth-supported templates. ” This statement is false. Recent studies have concluded that between the mucosa- and tooth-supported guides, there were no statistically significant differences for any of the outcome measures.1
- In the introduction section: “However, major inaccuracies in the implementation can usually be traced back to application errors and not to the process per se (14).” Revise this sentence.
- The author did not assess heterogeneity before meta-analysis. Firstly, the studies should have a very similar patient population and methodology. Secondly, heterogeneity should be measured using the chi-squared test and I2. When there is a high level of heterogeneity, it’s not appropriate to perform a meta-analysis.
- Which model did you use for the meta-analysis? Fixed effect or random-effects? Please add this information.
- Noun of the included studies compared the in vitro navigation accuracy with clinical navigation accuracy. It’s not appropriate to pool the data and do the descriptive analysis between the clinical and in-vitro studies.
- Please discuss the limitations of this systematic review.
References
- Gallardo R, Natali Y, Silva‐Olivio I R T, et al. Accuracy comparison of guided surgery for dental implants according to the tissue of support: a systematic review and meta‐analysis[J]. Clinical oral implants research, 2017, 28(5): 602-612.
Author Response
The systematic review provided by the authors is a work of a great interest for the journal. However, it is the revisor idea that some changes should be carried out in order to simplify some important passages. Therefore the authors are invited to carry out these major revisions outlined by the reviewer.
We would like to thank you for the assessment that the manuscript is a work of interest for the Journal of Clinical Medicine.
We have carefully reviewed the reviewer´s advice and incorporated inspired changes.
Since Reviewer 2 has shown a revision of the translation into English, we have had this done by the MDPI English editing service.
We hope that the manuscript will now have a quality that it can be published in the Journal of Clinical Medicine.
Response to Reviewer 2
This article systematically reviewed the previous researches investigating the accuracy of dynamic navigation dental implant surgery and made a meta-analysis of the pooled data. Overall, the topic lacks significance and there are some major flaws in the methodology. The detailed comments are as follows.
We would like to thank the reviewer for his effective comments, which will certainly increase the quality of the submitted manuscript.
We think that with the increasing number of systems for dynamic navigation and the increased use of these methods, it makes sense to systematically process the existing data. The number of systematic reviews and meta-analyses for static navigation proves the fundamental interest in computer-assisted methods.
- For the topic, the accuracy of the dental navigation system itself is not a question of much clinical significance. It’s more valuable to investigate whether dynamic navigation is superior to static computer guide systems or freehand in terms of accuracy, cost-effect, or patient-centered outcomes. Most of the included studies compared dynamic navigation with static guide or freehand. However, the author did not involve any analysis regarding those important comparisons in this systematic review.
We consider it important to look at the accuracy of this procedure. The clinical feasibility of these procedures requires sufficient accuracy. Thereview presented here will contribute to this. Of great clinical relevance is our statement that accuracy depends significantly on the system used.This differs in this form from static methods with drilling templates. The relevance of reviews to accuracy is also demonstrated by the large numberof systematic reviews for static navigation. Here, too, differentiated evaluations and comparisons with other methods were published only later.However, this also presupposes a certain quantity of primary literature.
This review therefore focuses on accuracy.
Since we generally share the comments you have made, our working group is currently working on further insights in preclinical and clinical studies.
- In the introduction section: “Implants that are placed using mucosa-supported templates show greater deviations from the planned implant position to the achieved implant position than those that were placed using tooth-supported templates. ” This statement is false. Recent studies have concluded that between the mucosa- and tooth-supported guides, there were no statistically significant differences for any of the outcome measures.1
Thank you for this valuable note. Unfortunately, we had overlooked this interesting review, so that a traditional prejudice can now be critically questioned. We have changed the section to:
“Various authors suspected that implants that are placed using mucosa-supported templates show greater deviations from the planned implant position to the achieved implant position than those that were placed using tooth-supported templates (10). Raico Gallardo et al. were able to refute this claim in their meta-analyse (14).”
- In the introduction section: “However, major inaccuracies in the implementation can usually be traced back to application errors and not to the process per se (14).” Revise this sentence.
We have corrected this sentence and explained it in more detail:
“However, major inaccuracies in the implementation implantation can usually be traced back to application errors and not to the process per se (15). In particular, errors in the positioning of the drilling template are to be mentioned here.“
- The author did not assess heterogeneity before meta-analysis. Firstly, the studies should have a very similar patient population and methodology. Secondly, heterogeneity should be measured using the chi-squared test and I2. When there is a high level of heterogeneity, it’s not appropriate to perform a meta-analysis.
That is an important point. Thank you for this comment.
In the forest plots, this analysis data is unfortunately only inconspicuously recorded. We have therefore included in the text:
“All forest plots confirmed a substantial heterogeneity in all parameter in both groups. The results of these analysis were P < 0.001 and I2 97.4% - 99.6%.“
- Which model did you use for the meta-analysis? Fixed effect or random-effects? Please add this information.
We have added:
“As there was evidence of hetereogeneity between the included studies, totals were calculated using random-effects meta-analysis for continuous variables. The models were based on the variances approach of DerSimonian and Laird. The heterogeneity was assessed using the Cochran Q test (P<0.001 (CI 95%) and I2 statistic (I2 > 50%)”
- Noun of the included studies compared the in vitro navigation accuracy with clinical navigation accuracy. It’s not appropriate to pool the data and do the descriptive analysis between the clinical and in-vitro studies.
We have approached this point very peripherally precisely for the reason of your restrictive statement. In order to provide truly meaningful data, clinical and in vitro data using the same technique and procedure would have to be available. However, there is insufficient primary literature for this. Our working group is currently working on such a comparison.
In early work on static navigation, clinical, cadaver and in vitro studies were even summarized in reviews. We wanted to avoid this methodological error.
Nevertheless, we wanted to give the reader an indication of whether in vitro studies show clinical relevance.
Wir haben zur Klarstellungfolgenden Satz eingefügt:
“To classify the in vitro studies on the clinical studies, we carried out a static analysis for their variance. This analysis should have a descriptivevalue, since there is not yet enough data on an actual one.”
- Please discuss the limitations of this systematic review.
We were very happy to take up this proposal and added a supplement to the discussion:
“The high degree of heterogeneity of the measured parameters suggests that the evaluated studies examined navigation systems with significantly different quality. The structure and arrangement of the marker structures seems to lead to different results. For reviews of static templated-guided implants, there is a lower heterogeneity (65). Nor can the present study make any statement on clinical suitability for practice.”
Round 2
Reviewer 1 Report
It is the review idea that the edited version of the manuscript is now suitable for publication.
Author Response
Thank you for the effort and time you have put into the review.
Thanks for the acceptance.
Reviewer 2 Report
The scientific terms used in this manuscript are confusing and not consistent with what was used in the previous studies. Please revise the terms as following:
Template-guided navigation revise to Static navigation / static guided surgery / static computer-assisted surgery
error revise to deviation
variance revise to deviation
deviation at entry point revise to Coronal deviation / deviation at implant platform
deviation at exit point revise to Apical deviation / deviation at implant apex
3D global deviation revise to Global deviation
2D linear lateral deviation revise to Lateral deviation
RESULTS SESSION
The structure of the RESULT section is wired. For accuracy of computer-assisted surgery, the three most important measurements are 1) coronal deviation, 2) apical deviation, 3)angular deviation. However, there are only two sessions in the current results: 3.3.1 coronal deviation and 3.3.2 angle variance.
The author made 5 forest plots for the following 5 measurements: 1. angular in vitro
- coronal global in vitro; 3. coronal lateral in vitro; 4. angular clinical; 5. coronal global clinical. This selection of measurements is quite wired. Firstly, there is only angular deviation and coronal deviation, while no apical deviation. Since the coronal lateral deviation is a part of the coronal global deviation, there is no need to make plots for both while no plots for apical deviation. Secondly, the authors choose 3 measurements from in vitro study, while only 2 measurements from the clinic study. I would suggest the author making plots for angular deviation, coronal global deviation, and apical global deviation.
In RESULTS section, 3.3.2, “The dynamic navigation also showeds comparable accuracies compared to the template-guided navigation.” Move this statemen to Discussion section, it’s not your study results.
DISCUSSION SESSION
In DISCUSSION section “In the present evaluation, the focus was on the angle deviation and the deviation at the implant exit point. This was justified in the prosthetic importance of these parameters. Inclined implant axes make it difficult to correctly design the approximal contacts.” I don’t see the rationale of only focusing on angular deviation and implant entrance point. The accuracy at the implant apex is equally important as coronal deviation and the angular deviation. The implant apex is usually adjacent to vital anatomical structures like the sinus, mental foramen, and mandible canal. Moreover, the apical deviation is larger than the coronal deviation. It’s important to know the apical deviation to assess the risk of injury vital structures.
CONCLUSION SESSION
Please also report deviation at the implant entrance.
It’s not appropriate to conclude that “These results are of the same order of magnitude that can be achieved with static navigation methods.” because you did not investigate this question in this study.
Delete “There are currently no studies that examine the practical implementation of dynamic navigation in more detail.”
Author Response
Reply to Review Report No.2 Reviewer 2
Thank you for the further comments in your review.
We were happy to take up these comments and amended the manuscript accordingly.
We now hope that the manuscript has also reached the quality of being published from your side.
All changes in revision 2 are set in this font color
The scientific terms used in this manuscript are confusing and not consistent with what was used in the previous studies. Please revise the terms as following:
Template-guided navigation revise to Static navigation / static guided surgery / static computer-assisted surgery
Page 2: we have changed in: … procedures for static guided implant placement …
Page 3: Table 1: in exclusion criteria we have changed in .. static guided implant placement ..
Page 14: we have changed in: .. compared to the static guided surgery.
and …meta-analysis for static guided implant placement….
and Table 6: …static navigation….
Page 15: we have changed in: …computer-assisted static guided implants…
error revise to deviation
Page 12 and 13:
Figure 4 and 7 we have changed in: … Forest plot demonstrating the angular deviation
(°) for all of the selected…
and Figure 5, 6 and 8: we have delated “error”. Now the sentences is: … Forest plot demonstrating the 3D global deviation (mm) at the entry point measured for all of the selected in vitro articles.
variance revise to deviation
Page 14: Subheading. We have changed in: Angle deviation
Table 6. We have changed in Angle Deviation
Here we used "entry point" and "exit point" synonymously.
This is really very confusing. Thank you for this very important point.
deviation at entry point revise to Coronal deviation / deviation at implant platform
Page 10: Table 4: we have changed in …deviation at the implant platform….
Page 11: Table 5: we have changed in …deviation at the implant platform….
Page 12: Figure 5: we have changed in … demonstrating the 3D global deviation (mm) at the implant
platform measured…
Page 13: Figure 6: we have changed in … demonstrating two-dimensional (2D) linear lateral deviation
(mm) at the implant platform measured…
Page 13: Figure 8: we have changed in … demonstrating the 3D global deviation (mm) at the implant
platform measured for all of the selected …
deviation at exit point revise to Apical deviation / deviation at implant apex
Page 1: Abstract: we have changed in: …deviation at the platform of the omplant ….
Page 3: Table1: we have changed in: …deviation at the implant platform...
Page 12: we have changed in: … given in five papers and the linear coronal deviation of the implant in
seven papers….
and: …. studies, data on both the angular deviation (Figure 7) and the global deviation at the
implant platform were available. (Figure (8)…
Page 15: we have change in: … the angle deviation and the deviation at the implant platform…
and: …The coronal position of the implant in particular has a decisive influence on the
esthetic result…
Page 16: we changed in: … in vitro were calculated at the platform of the implant…
3D global deviation revise to Global deviation
Page 1: Abstract, Page 3: Table 1, Page 4, Page 12, Page 12 Figure 5, Page 14, Page 14 Table 6,
2D linear lateral deviation revise to Lateral deviation
Page 12: Table 6, Page 13, Page 14,
we have changed “3D” in global and “2D” in lateral or we have delated “3D or 2D” in sentences like “3D global deviation” or “2D lateral deviation”
RESULTS SESSION
The structure of the RESULT section is wired. For accuracy of computer-assisted surgery, the three most important measurements are 1) coronal deviation, 2) apical deviation, 3)angular deviation. However, there are only two sessions in the current results: 3.3.1 coronal deviation and 3.3.2 angle variance.
We wanted to do this review without the apical deviation, as it is directly geometrically dependent on the angle and the implant length. You are, of course, quite right that these parameters are important when considering the risk of injury to anatomical structures. In addition, this measure is specified in all comparable static navigation studies.
Therefore, we have extended the manuscript:
3.3.2 Apical Deviation
The results of the meta-analysis showed comparable mean values. The global deviation in the clinical studies was 1.33 mm (95% CI, 0.98–1.68). In the in vitro studies, this global deviation was 1.04 mm (95% CI, 0.76–1.33). The forest plots demonstrate that the accuracy was highly system-dependent. Both evaluation groups showed outliers for the mean accuracy. It can also be seen from the graph that the scatter around the mean values in the clinical examinations is higher.
and: Forest plot 6new and 10new
and: a new column in table 6 with “Global Apical Deviation”
The author made 5 forest plots for the following 5 measurements: 1. angular in vitro
- coronal global in vitro; 3. coronal lateral in vitro; 4. angular clinical; 5. coronal global clinical. This selection of measurements is quite wired. Firstly, there is only angular deviation and coronal deviation, while no apical deviation. Since the coronal lateral deviation is a part of the coronal global deviation, there is no need to make plots for both while no plots for apical deviation. Secondly, the authors choose 3 measurements from in vitro study, while only 2 measurements from the clinic study. I would suggest the author making plots for angular deviation, coronal global deviation, and apical global deviation.
We have now included 7 forest plots in the manuscript:
Clinical: angle deviation, global deviation at the implant platform and global deviation at the implant apex.
In vitro: the same measurements: angle deviation, global deviation at the implant platform and global deviation at the implant apex.
In addition, we included the linear lateral deviation as a further measurement in the in vitro evaluation, since the global coronal deviation is not specified in the older studies.
The editor should please decide whether this evaluation should appear as a supplement.
In RESULTS section, 3.3.2, “The dynamic navigation also showeds comparable accuracies compared to the template-guided navigation.” Move this statemen to Discussion section, it’s not your study results.
We deleted this sentence in the "Results" section.
This statement is already described in the discussion (“The results of this meta-analysis are in the range of the results obtained from meta-analyses for static navigation [10,65,67]”).
Therefore, we have not added any further additions here.
DISCUSSION SESSION
In DISCUSSION section “In the present evaluation, the focus was on the angle deviation and the deviation at the implant exit point. This was justified in the prosthetic importance of these parameters. Inclined implant axes make it difficult to correctly design the approximal contacts.” I don’t see the rationale of only focusing on angular deviation and implant entrance point. The accuracy at the implant apex is equally important as coronal deviation and the angular deviation. The implant apex is usually adjacent to vital anatomical structures like the sinus, mental foramen, and mandible canal. Moreover, the apical deviation is larger than the coronal deviation. It’s important to know the apical deviation to assess the risk of injury vital structures.
Page 15: we added: “However, in order to evaluate the possible risk of injury sensitive anatomical structures such as the sinus, the mental foramen, and the mandible canal, these values were also evaluated.
CONCLUSION SESSION
Please also report deviation at the implant entrance.
Page 16: we have added: The mean global deviation at the implant apex was 1.33 mm clinically and 1.04 mm in vitro.
It’s not appropriate to conclude that “These results are of the same order of magnitude that can be achieved with static navigation methods.” because you did not investigate this question in this study.
Yes, this is correct, that we can only make statements about dynamic navigation from our own work.
Therefore, we have changed the sentence in:
“These results of dynamic navigation show similar values as described in various systematic reviews for static navigation.”
Delete “There are currently no studies that examine the practical implementation of dynamic navigation in more detail.”
We have deleted this sentence.